# *Candidiasis* and Mechanisms of Antifungal Resistance

**DOI:** 10.3390/antibiotics9060312

**Published:** 2020-06-09

**Authors:** Somanon Bhattacharya, Sutthichai Sae-Tia, Bettina C. Fries

**Affiliations:** 1Division of Infectious Diseases, Department of Medicine, Stony Brook University, Stony Brook, New York, NY 11794, USA; sutthichai.sae-tia@stonybrookmedicine.edu (S.S.-T.); Bettina.Fries@stonybrookmedicine.edu (B.C.F.); 2Department of Molecular Genetics and Microbiology, Stony Brook University, Stony Brook, New York, NY 11794, USA; 3Veterans Administration Medical Center, Northport, New York, NY 11768, USA

**Keywords:** candidiasis, antifungal resistance, azole resistance, efflux pump, ergosterol biosynthesis, echinochandin resistance, polyene resistance, 5-Flucytosine resistance

## Abstract

*Candidiasis* can be present as a cutaneous, mucosal or deep-seated organ infection, which is caused by more than 20 types of *Candida* sp., with *C. albicans* being the most common. These are pathogenic yeast and are usually present in the normal microbiome. High-risk individuals are patients of human immunodeficiency virus/acquired immunodeficiency syndrome (HIV/AIDS), organ transplant, and diabetes. During infection, pathogens can adhere to complement receptors and various extracellular matrix proteins in the oral and vaginal cavity. Oral and vaginal *Candidiasis* results from the overgrowth of *Candida* sp. in the hosts, causing penetration of the oral and vaginal tissues. Symptoms include white patches in the mouth, tongue, throat, and itchiness or burning of genitalia. Diagnosis involves visual examination, microscopic analysis, or culturing. These infections are treated with a variety of antifungals that target different biosynthetic pathways of the pathogen. For example, echinochandins target cell wall biosynthesis, while allylamines, azoles, and morpholines target ergosterol biosynthesis, and 5-Flucytosine (5FC) targets nucleic acid biosynthesis. Azoles are commonly used in therapeutics, however, because of its fungistatic nature, *Candida* sp. evolve azole resistance. Besides azoles, *Candida* sp. also acquire resistance to polyenes, echinochandins, and 5FC. This review discusses, in detail, the drug resistance mechanisms adapted by *Candida* sp.

## 1. Introduction

*Candida* species are commensals and thus are part of the normal human flora and are localized on skin and gastrointestinal and genital tracts. However, *Candida* can also cause various infections in susceptible patients that includes elderly, hospitalized, or immunosuppressed patients. Invasive *Candida* infection is one of the most common fungal infections globally [1]. In the United States, *Candida* sp. were reported to be one of the leading causes of healthcare-associated infections [1]. Amongst the different *Candida* sp., *Candida albicans* is the most commonly recovered (37%) from clinical species, followed by *Candida glabrata* (27%). Other clinically relevant species recovered from blood stream infections include *C. parapsilosis* (14%), *C. krusei* (2%), *C. tropicalis* (8%), *C. dubliniensis* (2%), *C. lusitaniae* (2%), and the most recent, *C. auris*. *C. auris* is an emergent multi-drug-resistant pathogen that is often misidentified and at present is a major concern in healthcare settings. *C. auris* reported cases increased by 318% between 2015 and 2018. As per the Centers for Disease Control and Prevention (CDC), an estimated 34,000 cases of *Candidiasis* were reported in hospitalized patients and about 1700 people died in 2017 [2]. *Candidiasis* has a diverse clinical spectrum ranging from non-life threatening superficial mucocutaneous infections to devastating invasive disease associated with candidemia. In fact, the attributable mortality observed with candidemia is between 30% and 47% [3]. *Candida* infection is also commonly associated with medical devices such as central venous catheters, cardiovascular devices, and urinary catheters [4]. An episode of candidemia can lead to seeding of any organs, including the liver, spleen, bones, joints, eyes, or brain.

Because of a lack of rapid diagnostic assays for invasive *Candidiasis*, most *Candidiasis* cases are still diagnosed by routine fungal cultures of blood, urine tissue, and other body fluids. This method can have a low sensitivity, and in some cases, it can also render false-positive results because of contamination, which can occur in the process. Commonly, empiric anti-fungal therapy is initiated in febrile or septic patients in the intensive care unit with an indwelling central venous catheter, recent abdominal surgery, or chemotherapy in the absence of response to anti-microbial therapy. This approach can lead to the unnecessary use of antifungal agents and promote the emergence of resistance in individuals without invasive *Candidiasis* or a delay in effective antifungal therapy for those who are infected.

The strategy for treatment of invasive *Candidiasis* depends on the patient’s immune status, location, and severity of the infection. In addition to an adequate source control, removal of infected medical devices and antifungal agents have been important therapeutic tools for invasive *Candida* infections [5]. At present, four main classes of antifungal drugs with activity against *Candida* species are available, including polyenes, azoles, echinocandins, and 5-Flucytosine (5FC). Similar to antibiotics for bacterial infection, an emergence of antifungal resistance among *Candida* species is a serious threat to public health worldwide. According to the US Center of Disease Control and Prevention (CDC) 2019 report of antibiotic resistance threat, more than 34,000 cases and 1700 deaths annually were due to drug-resistant *Candida* sp. In addition, 323 cases of emerging multidrug- resistant *Candida auris* infection were reported.

In the current review, we discuss the molecular mechanisms of action of these antifungal agents as well as mechanisms of drug resistance used by *Candida* sp.

## 2. Antifungals and Their Targets

*Candidiasis* can be treated with antifungals that belong to different classes of drugs and target different cellular processes, thereby either inhibiting (fungistatic) or killing (fungicidal) the growth of this pathogenic yeast. These cellular processes include the biosynthesis of the cell wall, cell membrane, and biosynthesis of RNA. Each of these biosynthesis processes involves a series of enzymes. The targets and the mechanisms of antifungals employed for treating *Candidiasis* are outlined below.

### 2.1. Antifungals that Target Ergosterol and Its Biosynthesis

Ergosterol is the major sterol component of fungal cell membranes, including the plasma and mitochondrial membranes. It is vital for fungi to maintain the structure and function of these membranes. Together, sterols and sphingolipids form lipid rafts in the cell membrane. These lipid rafts contain many biologically important proteins, which are involved in signaling, response to stress, mating, and nutrient transport [6,7]. Ergosterol biosynthesis is catalyzed by a cascade of 25 different enzymes (Figure 1A). This biosynthetic pathway constitutes an ideal target for drugs because ergosterol is a very important lipid for fungi and plants but is not present in humans. Several drugs target either the biosynthesis of this lipid or ergosterol itself, and are described below (Figure 1).

**Azoles:** Azoles are five-membered heterocyclic compounds with antifungal properties. They are broadly classified into two groups that include imidazole and triazole. Imidazoles contain two nitrogen in the azole ring and include antifungals clotrimazole, econazole, ketoconazole, miconazole, and tioconazole. Triazoles contain three nitrogen in the azole ring and include fluconazole, itraconazole, voriconazole, isavuconazole, and posaconazole. Fluconazole is the most common azole used during therapy. Second generation of triazoles that include voriconazole, posaconazole, and isavuconazole are more potent against resistant pathogens. Isavuconazole is a novel azole, as effective as voriconazole but less toxic than voriconazole. Toxicity is lower in isavuconazole, since it is highly water soluble and does not require beta-cyclodextrin in its intravenous (IV) formulation, which is required for voriconazole, causing reduced nephrotoxicity [8,9]. Isavuconazole can be administered both intravenously (IV) and orally and its toxicity profile is similar to that of fluconazole but is more active [10,11].

Azoles are the most common antifungal drug class used for treating and preventing *Candida* infections. Azoles target the enzyme 14α–demethylase (Erg11p), an important enzyme in ergosterol biosynthesis (Figure 1) [12]. Azoles bind to Erg11p, thereby effectively lowering the ergosterol levels of the cell. When Erg11p is inhibited, other enzymes in the pathway (Erg6p, Erg25p, Erg26p, Erg27p, and Erg3p) synthesize a fungistatic toxic sterol (14α methylergosta 8-24 (28) dienol, Figure 1B) [7]. Besides this, azoles are also responsible for elevating the levels of reactive oxygen species (ROS) [13]. Both the elevated ROS levels and toxic sterol production inhibit the growth of the infecting fungus.

**Polyenes:** This class of drugs target ergosterol in the plasma membrane and are fungicidal. They bind to ergosterol and form pores [14]. Pore formation causes rapid leakage of monovalent ions (K+, Na+, H+, and Cl−) and subsequent fungal cell death. Polyene drugs include amphotericin B and nystatin, but only amphotericin B is used for systemic treatment. Ergosterol is more sensitive to amphotericin B than cholesterol, the common mammalian sterol. To decrease its toxicity, the conventional amphotericin B, which is complexed with sodium deoxycholate (ABD), has been modified as a cholesteryl sulfate complex (ABCD), as a lipid complex (ABLC), and as a liposomal formulation (LAMB). These formulations may exhibit different pharmacokinetic characteristics compared to conventional amphotericin B [15,16]. For the treatment of invasive *Candidiasis*, amphotericin B is rarely required. This drug is mostly chosen when the *Candida sp*. is resistant to other drug classes or the drugs do not penetrate into the relevant niche.

Other antifungals that inhibit growth of a large variety of fungi but are not used to treat *Candidiasis* are listed below:

**Allylamines:** These antifungals target Squalene epoxidase (Erg1p) in the ergosterol biosynthesis pathways [7]. These drugs include terbinafine (Lamisil), flunarizine, and naftifine. Terbinafine (Lamisil) is commonly used in treating dermatophyte infections [17].

**Morpholines:** This class of drugs include fenpropimorph, tridemorph, and amorpholine. They target the ergosterol biosynthetic enzyme C-14 sterol reductase (Erg24p) [7]. The morpholines are commonly used in agriculture, and exhibit high toxicity in humans [17]. Only 5% amorolfine hydrochloride containing nail lacquer solution is used for treating nail dermatophyte infections [18].

### 2.2. Inhibitors of Cell Wall Biosynthesis

Besides ergosterol, cell wall biosynthesis of *Candida* sp. is also targeted by various antifungals. The cell wall is the rigid outermost layer of the fungal cell and is the first line of defense, protecting the fungal cells from osmotic stress. Mammalian cells do not have cell walls and therefore the enzymes of the cell wall biosynthetic pathways are important antifungal targets [19]. Echinocandins antifungal drugs target the cell wall and include caspofungin, micafungin, and anidulafungin. They target β1-3 glucan synthase enzyme, which is encoded by three genes *FKS1*, *FKS2*, and *FKS3* [20]. β1-3 glucan synthase enzyme is a complex of three proteins, Fks1p, Fks2p, and Fks3p, that uses Uridine Diphosphate Glucose (UDP-glucose) to synthesize β1-3 glucan, an important component in the fungal cell wall (Figure 2) [21]. These drugs are usually fungicidal and commonly chosen because they exhibit low toxicities for humans [22]. Echinochandins are usually administered to patients with moderately severe to severe illness or to patients with prior exposure to azoles [23].

### 2.3. Inhibitors of Nucleic Acid Biosynthesis

Other important antifungals are those that target nucleic acid biosynthesis. These were originally designed in the 1950s and are one of the oldest classes of antifungal drugs. 5-Flucytosine (5FC) is an antifungal that interferes with nucleic acid biosynthesis. The susceptible cells import 5FC via the cytosine permease enzyme [24]. 5FC is converted to 5-Fluorouracil (5FU), which is metabolized to 5-Fluorouridine triphosphate (5FUTP). 5FUTP is incorporated in the fungal RNA instead of uridine triphosphate (UTP), thereby affecting protein translation. Alternatively, 5FU can be converted to 5-Fuorodeoxyuridine monophosphate (5FdUMP) that inhibits thymidylate synthase, an important enzyme in DNA biosynthesis [24]. The pathway is summarized in Figure 3.

## 3. Drug Efficacies and Therapeutic Failure

Several important factors may contribute to treatment failure. These are listed below.

**Bioavailability:** Differences in drug bioavailability between different tissues can contribute to the effectiveness of antifungal agents in treating fungal infections. For example, bioavailability of azole drugs in vaginal tissues with a low pH is much lower than that in blood [25]. Also, infections in the central nervous system are particularly difficult to treat because echinocandins do not cross the blood–brain barrier sufficiently and therefore, adequate drug levels are not achieved. Furthermore, echinocandins are bound to albumin and are not excreted into the urine. Therefore, they may not penetrate at high levels to adequately treat *Candida* infections in the urinary tract. Importantly, different lipid formulations vary in their tissue distribution and toxicity. Specifically, although LAMB has a better overall therapeutic index, ABLC achieves higher lung and kidney tissue concentrations [16].

**Host Immune system:** The host immune system plays a critical role in successful therapeutics. Especially, static antifungal drugs such as the azoles require support from the host’s immune system for successful treatment of invasive *Candida* infections. In severely neutropenic patients, these drugs may be less effective and supportive treatment with growth factors to boost engraftment is required.

Successful azole therapy depends on the drug and on the host’s immune response to the fungus. Without a functional immune system, azoles cannot control the fungal infections. For example, *Candida* sp. can colonize the mouth of 64% to 84% of human immunodeficiency virus HIV-infected patients [26]. Development of azole-resistance in these patients has many contributing factors. Typically, oral candidiasis regularly relapses in HIV-infected patients with low CD4^+^ cell counts (<50/mm^3^) [27]. These patients typically receive a long-term, low-dose azole therapy, which can select for azole-resistant *Candida* sp. In one case, a series of 17 *C. albicans* clinical isolates were obtained from a single HIV patient receiving azole therapy over two years. Azole-resistance increased steadily as these isolates were exposed to increasing amounts of drugs [28].

**Biofilm formation:** Fungi can develop biofilms on the surface of medical equipment, such as catheters. These biofilms are not only resistant to the penetration of some antifungals but also the drug targets may be less expressed [29].

**Drug pharmacokinetics, infection severity and side-effects:** The pharmacokinetics of the drugs is an important factor that contribute to drug efficacies. These include drug metabolism, absorption, and distribution that can alter drug effectiveness. Additionally, drug efficacy can depend on the severity of the infection, and the population size of the infecting organisms [26]. Further, while administering an antifungal, one should consider its side effects. For example, 5FC was used in the past as monotherapy for treating candidiasis, aspergillosis, and cryptococcosis. Side effects of this drug are substantial and include, colitis, bone-marrow suppression, and liver toxicity. These drugs are now used mostly in combination with azoles for complicated fungal infections or in cases of infections with multidrug-resistant *Candida* sp. [24]. Toxicity is also a common side effect of amphotericin B [17]. Nephrotoxicity is commonly associated with amphotericin B that is caused by renal tubular acidosis, azotaemia, impaired renal concentrating ability, and electrolyte abnormalities [30,31]. Amphotericin B promotes inflammatory cytokines release by a Toll-like receptor- and CD14-dependent mechanism, which leads to systemic side effects (fever, nausea, hypotension, and shaking) [32]. Azoles and echinochandins have relatively low side effects. For example, in rare circumstances, long-term use of azoles, especially posaconazole and voriconazole, can cause liver injury [33]. Similarly, echinochandins also show minimum side effects, which include nausea, vomiting, and abdominal pain [34].

**Antifungal drug resistance:** Another important factor that can cause therapeutic failure is antifungal drug resistance of the infecting *Candida* sp. For example, prolonged use of a fungistatic azole drugs like fluconazole may cause the pathogenic yeasts to develop resistance, making the drug less effective. Fluconazole is the most common azole used for prophylaxis and treatment of *Candida* infections. However, several *Candida* sp. have evolved resistance to azoles, which is an emerging problem causing therapeutic failures [17]. For example, azole resistance in *C. albicans* became a severe problem in the 1990s, when 90% of acquired immunodeficiency syndrome (AIDS) patients had oral candidiasis, many received a long-term azole therapy and certain patient populations developed resistance [26].

## 4. Antifungal Drug Resistance

### 4.1. Azole Drug Resistance

For decades, extensive research has analyzed the molecular mechanisms of azole resistance in pathogenic fungi. The known mechanisms are described below and summarized in Figure 4.

The different mechanisms of azole resistance are described below. These mechanisms are all well studied, are all equally important, and are frequently observed in several drug-resistant *Candida* clinical isolates.

#### 4.1.1. Over-Expression of Membrane Transporters

A large number of membrane proteins are present in pathogenic yeast. These membrane proteins are present in the cell membrane [35], vacuolar membrane [36], and mitochondrial membrane [37]. They play various physiological roles that include environmental sensing, nutrient transport, signal transduction, drug efflux, drug modification, and drug detoxification [35]. For example, mitochondrial membrane-localized protein Atm1p, an ABC transporter, plays an important role in iron homeostasis [38], while vacuolar membrane-localized membrane transporter Mlt1p transports phosphatidyl choline (PC) [36]. Further, a single membrane transporter can have multiple physiological roles. For example, a recent study demonstrated that Mlt1p transporter is responsible for transporting PC as well as importing azoles into vacuoles. Deletion of Mlt1p caused susceptibilities to methotrexate and azoles in *C. albicans* [36].

Two types of membrane transporters have been identified that correlate to azole resistance in fungi. They are described below (Figure 4 Grey box 1):**a)** **ABC-transporters.** Adenosine Triphosphate (ATP)-binding cassette transporters (ABC-T) are active transporters requiring ATP as an energy source. Each ABC-T consists of two membrane-spanning domains (MSD), each containing six transmembrane segments, and two nucleotide binding domains (NBD). Each NBD consists of an ATP-binding cassette (ABC) that binds ATP [39]. Azoles are the substrates for the ABC-Ts that are listed in Table 1 [40].**b)** **MFS-transporters.** Major facilitator transporters (MFS-T) require a proton gradient of the plasma membrane as an energy source to transport xenobiotics. MFS-Ts do not have the NBDs that are characteristic of ABC-Ts, and they have 12 to 14 transmembrane segments [41]. Azoles are also the substrates of the MFS-Ts listed in Table 1.

In pathogenic fungi, increased expression of membrane transporters correlates with azole resistance. For example, overexpression of *CaCDR1*, *CaCDR2*, and *CaMDR1* is commonly observed in azole-resistant oral, systemic, and vaginal *C. albicans* clinical isolates [25,42]. In vitro, the disruption of genes encoding these transporters causes hyper-susceptibility to specific antifungals, including azoles [43].

While molecular mechanisms can be characterized into randomly collected resistant clinical isolates, matched isolates provide a more specific analysis of drug resistance. One matched set of 16 *C. albicans* clinical isolates was obtained from one individual over a two-year period of antifungal therapy. Increased expressions of *CaCDR1*, *CaCDR2*, and *CaMDR1* were observed in the matched resistant isolates when compared to their susceptible partners [25,28]. Analysis of random, non-matched resistant *C. albicans* isolates also shows increased efflux pump gene expression [25,42].

Besides *CaCDR1*, *CaCDR2*, and *CaMDR1*, increased expression of *CaFLU1* (an MFS-T) and *CaPDR16* (an ABC-T) has been observed in azole-resistant isolates [44]. CaPdr16p is a phosphatidylinositol transfer protein, disruption of which causes increased susceptibilities to various antifungals [45]. CaFlu1p is not considered a major azole transporter because in vitro overexpression does not confirm azole resistance [46]. The *C. albicans* genome contains many other membrane transporters. However, to date, none have been shown to effect azole resistance [47].

The efflux pumps *CaCDR1* and *CaCDR2* are regulated by transcription factor CaTac1p, while *CaMDR1* is regulated by transcription factor CaMrr1p [48,49]. Both transcription factors are zinc-cluster transcription factors (Zn_2_-Cys_6_). Gain of function (GOF) mutations in *CaTAC1* and in *CaMRR1* result in constitutive overexpression of their respective pumps, and azole resistance in many clinical isolates (Figure 4, grey box 1). Specifically, several GOF mutations in *CaTAC1*, including T225A, V736A, N972D, N977D, G980E, and G980W, have been correlated with resistance in clinical isolates [50]. GOF mutations in *CaMRR1*, including P683S and P683H, have also been correlated with resistance in clinical isolates [51,52]. GOFs in *MRR1* are also observed in *C. dubliniensis* isolates resistant to fluconazole. These mutations include T374I, S595Y, C866Y, T965Δ, and (D987-I998)Δ [53].

There are other transcription factors that regulate the expression of *CaCDR1* and *CaCDR2*, including the transcription factors, *CaNDT80*, *CaFCR1*, and *CaFCR3* [54,55]. To date, no studies have shown the association of *CaNDT80*, *CaFCR1*, and *CaFCR3* with resistance in clinical isolates. *CaMDR1* is transcriptionally regulated by *CaMRR1* as well as *CaCAP1* and *CaMCM1* [56]. In vitro, a truncated C-terminus of CaCap1p correlated with increased azole resistance [57].

In *C. glabrata*, overexpression of the ABC-T efflux pumps *CgCDR1*, *CgSNQ2*, and *CgPDH1* has been shown to contribute to azole resistance in clinical isolates [58,59]. Overexpression or disruption of ABC-T encoded by *CgPDR16* has also been associated with the azole resistance mechanism in *C. glabrata* [60,61]. Amongst the MFS-Ts, increased expression of *CgFLR1* and *CgQDR2* is also observed in azole-resistant clinical isolates [62,63].

GOF mutations in the transcription factor CgPdr1p activates CgPdr1p, which in turn induces the ABC-T pumps *CgCDR1*, *CgSNQ2*, and *CgPDH1* [64]. These GOFs include L328F, R376W, D1082G, T588A, T607S, E1083Q, Y584C, D876Y, L280F, N691D, S316I, D261G, R293I, R592S, G583S, S343F, and R376G. These GOFs significantly increased resistance to fluconazole [64]. GOF mutations in *CgPDR1* also regulate virulence. For example, GOF L280F in *PDR1* regulates adherence to macrophages and macrophage-mediated phagocytosis [65]. GOF in *CgPDR1* results in increased expression of adhesin proteins like Epa1, causing increased adherence [66]. *CgPDR1* also regulates MFS-T *CgQDR2*, which upregulates in response to azoles [67]. This upregulation of *CgQDR2* is caused by GOF L946S in *CgPDR1.* Deletion of *CgQDR2* showed susceptibility to azole, miconazole, suggesting the role of *CgQDR2* in azole resistance [65].

Mitochondria also plays an important role in efflux pump-mediated azole resistance in *C. glabrata*. Loss of mitochondria leads to azole resistance, which correlated with upregulation of *CgCDR1*, *CgPDH1*, and other genes [59].

Membrane transporter overexpression can also affect intrinsically azole-resistant *C. krusei*. In clinical isolates of *C. krusei*, overexpression of the ABC-T pumps CkAbc1p and CkAbc2p correlates with drug resistance, expression of both pumps can be induced by azoles, and expression of the pumps in a hyper-susceptible *S. cerevisiae* strain results in resistance [68]. However, azole resistance was not affected by efflux pump inhibitors, suggesting that efflux-mediated resistance is not due to efflux pumps alone in *C. krusei* [69]. Increased efflux pump expression also correlates with increased azole resistance in *C. parapsilosis* and *C. dubliniensis* [70,71].

Membrane transporter-based azole resistance was also recently observed in *C. auris*, an emerging multidrug-resistant fungal pathogen [72,73]. *C. auris* contains *C. albicans* homolog *CDR1*, which was found to be overexpressed in generationally aged *C. auris* that showed increased tolerance to fluconazole [72]. Recently, it was observed that increased expression of *C. albicans* homolog of *CDR1* contributed in the evolution of azole-resistant *C. auris* isolates [74]. This increased expression correlated with increased copy number of the transcription factor *TAC1* in the evolved strain when compared to its parent [74].

#### 4.1.2. Altered Ergosterol Biosynthesis

Besides membrane transporters, azole resistance via altered ergosterol biosynthesis was also demonstrated previously. These are summarized below.Mutation and/or overexpression of ergosterol pathway genes (Figure 4; Grey box 3)

***ERG11/CYP51*:** Erg11p is a cytochrome p450 enzyme that regulates a rate-limiting step in the ergosterol biosynthetic pathway [12]. It is an essential enzyme in the pathway, and cellular ergosterol levels decrease when Erg11p is inhibited by azoles [75]. *ERG11* overexpression is linked with azole resistance in many fungi. Many azole-resistant *C. albicans* clinical isolates show increased *CaERG11* expression [25,42,76]. *CgERG11* overexpression is observed in azole-resistant *C. glabrata* isolates [77]. Overexpression of *ERG11* was also recently observed in generationally aging *C. auris* that showed increased resilience to azoles [72]. Several point mutations in *ERG11* have been identified in resistant clinical isolates. For example, CaErg11p mutations A61V, A114S, Y132F, Y132H, K143Q, K143R, Y257H, S405F, G448E, F449S, G464S, R467K, and I471T contribute to azole resistance in *C. albicans* [78]. Many of these point mutations lower azole binding in the active site of the enzyme. Point mutations in *ERG11* associated with resistance have been identified in other azole-resistant *Candida* sp. These include, *C. glabrata* mutations C108G, C423T, and A1581G, and *C. krusei* mutations A497C and G1570A [79].***ERG3*:** Besides *ERG11*, *ERG3* is also linked with azole resistance. Erg3p is a C5 sterol desaturase enzyme, and converts episterol to ergosta-5,7,24 (28)- trienol during ergosterol biosynthesis (Figure 4; Grey box 4). When Erg11p is inhibited, Erg3p and Erg6p synthesize a toxic sterol, 14α methyl ergosta 8,24 (28)-dien-3β, 6α-diol, in both *C. albicans* and *C. glabrata*, [80,81,82]. Disruption of *ERG3* results in increased azole resistance in *C. albicans* [83]. Q139A in Erg3p is reported in azole-resistant *C. glabrata* isolates [84].***ERG6*:** In the presence of azoles, *ERG6* contributes to the formation of the toxic diol from lanosterol [80,81,82] (Figure 4; Grey box 5). Erg6p is a Δ24 sterol C-methyl transferase, a non-essential enzyme in the ergosterol biosynthetic pathway. Significant azole resistance is observed in the heterozygous *ERG6* deletion in *C. albicans* [85]. Similarly, Δ*erg6* in *S. cerevisiae* showed increased azole resistance [86], increased membrane permeability, and low Pdr5p efflux activity [87]. This suggests that *ERG6*-dependent azole resistance is the result of toxic sterol formation, and not due to efflux pump overexpression.

Gain of Function Mutation (GOF) in UPC2 (Figure 4; Grey box 2)

Upc2p is a transcription factor that regulates the majority of ergosterol biosynthetic genes. For example, it regulates the expression of *CaERG2* and *CaERG11* in *C. albicans* [88]. Upc2p is auto-regulatory [89], and is induced by azoles, anaerobic growth, and low levels of ergosterol [90]. Upc2p is a Zn_2_-Cys_6_ zinc cluster transcription factor [88], and is well characterized in *C. albicans.* The N-terminus of Upc2p is the DNA binding domain, while the C-terminus is an activation/regulatory domain [91]. Ergosterol binds to the C-terminus of Upc2p and negatively regulates the transcription of *UPC2* [89]. Several mutations in CaUpc2p, including G648D, G648S, A643T, A643V, Y642F, G304R, A646V, and W478C are described in azole-resistant clinical isolates [76]. Seven of these mutations increased the expression of *CaERG11* [76]. Besides *CaERG11*, GOF mutations in *CaUPC2* increase the expression of other ergosterol pathway genes. For example, A643V GOF mutation in *CaUPC2* induces the expression of *CaERG2*, *CaERG3*, *CaERG5*, *CaERG6*, *CaERG9*, and *CaERG10* [92]. However, this mutation only causes a two-fold increase in azole resistance [92]. In vitro, GOF mutations A643T and A648D in *CaUPC2* significantly increased azole resistance [93]. Since *C. albicans* is diploid, GOF mutations in both *CaUPC2* alleles will cause more resistance than a GOF mutation in one allele [93].

The *C. glabrata* genome contains two homologs of *CaUPC2*, *CgUPC2A* and *CgUPC2B* [94]. *CgUPC2A* is an important regulator of the ergosterol pathway in *C. glabrata*, and is required for azole resistance [94]. Deletion of *CgUPC2A* in *C. glabrata* leads to azole susceptibility, whereas no effect is observed in the *CgUPC2B* deleted strain [94]. To date, no GOF mutations have been reported in *CgUPC2A* or *CgUPC2B.*

#### 4.1.3. Altered Sterol Import

Sterol import as a potential mechanism of azole resistance has been identified recently (Figure 4 Grey box 6). Azoles lower the ergosterol levels of the cell, which can be compensated by exogenous sterol import. Sterol import has been well characterized in *S. cerevisiae*, *C. albicans*, and *C. glabrata* [95]. Both *S. cerevisiae* and *C. glabrata* import sterols under anaerobic or microaerophilic conditions using the sterol importers Aus1p and Pdr11p [96,97,98], and show increased azole resistance [95]. Mutations in *UPC2*, as well as defects in the biosynthesis of ergosterol, and heme, can cause increased sterol import [99,100]. For example, *S. cerevisiae* strains with mutated *ScERG1* and *ScERG7* have increased sterol import [101].

Unlike *C. glabrata* and *S. cerevisiae*, *C. albicans* imports sterols aerobically, increasing azole resistance in the presence of both serum and cholesterol [95]. Thus, *C. albicans* may develop resistance by importing cholesterol and serum from the blood.

#### 4.1.4. Genome Plasticity

Genomic variations including loss of heterozygosity (LOH) and aneuploidy can correlate with azole resistance in fungi. If one allele of a gene is mutated, LOH can copy the mutation to the second allele. In clinical isolates of *C. albicans*, LOH has been observed in *CaTAC1*, *CaERG11*, and *CaMRR1*, correlating with increased resistance [102]. Clinical isolates of *C. albicans* can also include segmental aneuploidy, in which two copies of the left arm of chromosome 5 containing *CaERG11* and *CaTAC1* form an isochromosome that correlates with azole resistance [103]. Additionally, trisomy in chromosome 4 also caused an increase in azole resistance [104]. However, this observed increased resistance was neither attributed to increased efflux pump activities nor attributed to altered ergosterol biosynthesis in the clinical isolate. Thus, the azole resistance mechanisms by chromosome 4 trisomy are still unknown and need to be studied in the future. In another study, loss of one homologue of chromosome 4 contributed significantly with increased azole resistance [105]. Further, trisomy in chromosome 3 and chromosome R in *C. albicans* correlates with increased triazole resistance [105,106]. Trisomy of chromosome 3 develops under prolonged azole exposure. Chromosome 3 hosts efflux pump encoding genes *CDR1* and *CDR2* [105], and increased chromosome 3 copy number caused increased *CDR1* and *CDR2* expression [105]. Aneuploidy in another chromosome, chromosome 6, also corelates with azole resistance [105]. Chromosome 6 hosts efflux pump-encoding gene *MDR1* and increased expression of *MDR1* corelated with chromosome 6 trisomy under fluconazole exposure [105].

Chromosomal alterations are also observed in *C. glabrata*. For example, segmental rearrangements were previously observed in chromosome M and F in azole-resistant *C. glabrata* clinical isolates. Chromosome M harbors efflux pump-encoding gene *CgCDR1*, which was duplicated in these isolates, while chromosome F harbors efflux pump-encoding gene *CgPDH1* [107]. Besides aneuploidy, alteration in gene copy numbers is also associated with increased azole resistance. For example, increased copy number of *ERG11* in *C. glabrata* caused increased azole resistance [108]. Recently, duplication in genes *CDR1* and *ERG11* was also observed in generationally aging *C. auris* cells that rendered increased resilience to fluconazole [72].

#### 4.1.5. Other Hypothetical Azole Resistance Mechanism—Altered Azole Import

Several studies have hypothesized that defective azole import may contribute to drug resistance (Figure 4 Grey box 7). Azoles are imported by facilitative diffusion (FD) in *C*. *albicans* [109], and the import is energy-independent. Clinical isolates of many pathogenic fungi vary in azole import [109]. However, to date, no correlation has been observed between azole import and resistance. Further studies are required to identify the azole importers and characterize their roles in azole resistance.

### 4.2. Resistance to Other Drugs

Polyene resistance is also linked to changes in both *ERG3* and *ERG6*. For example, in vitro, disruption of *ERG3* and *ERG6* causes decreased ergosterol levels, and amphotericin B resistance in *C. albicans* and *C. glabrata* [110]. However, polyene resistance in clinical isolates has not been well characterized.

Echinocandins are the newest antifungal category, targeting β 1-3 glucan synthase, an enzyme important for cell wall biosynthesis localized to the plasma membrane [111]. The enzyme is encoded by the homologs *CaFKS1* and *CaFKS2*. Mutations in *CaFKS1* are observed in echinocandin-resistant clinical isolates of *C. albicans* [111], mutations clustered in two regions (amino acid regions 637–654 and 1345–1365) [22,112]. Mutations in *CaFKS2* can cause echinocandin resistance in *C. albicans* in vitro, but has not been observed in clinical isolates [113]. *FKS1* point mutations have also been observed in *C. glabrata*- and *C. krusei*-resistant isolates [113]. *FKS1* and *FKS2* point mutations spontaneously arise in the presence of echinocandin selection pressure in *C. glabrata*, with *FKS2* point mutation outnumbering *FKS1* point mutation by 2 to 1 [113,114]. *FKS1* point mutations involved in echinocandin resistance include S629P, F625Δ, and F625C. *FKS2* point mutations associated with increased echinocandin resistance are F659Δ, S663F, R1378S, R1378G, S663P, P667H, P667T, E655G, and E655K [113,114]. Besides point mutations in *FKS*, genome plasticity also causes increased echinocandin resistance. For example, chromosome 2 trisomy in *C. albicans* causes caspofungin resistance [115]. However, the specific mechanism behind this needs to be studied.

The other medically important drug 5FC is metabolized in the pyrimidine salvage pathway [116], and is used in treating candidiasis in combination with other antifungals. 5FC resistance is observed in *Candida* sp. In *C. albicans.*, 5FC is imported by cytosine permease, CaFcy2p, and is then deaminated to 5FU by cytosine deaminase, CaFcy1p. 5FU is then converted to 5FUMP by phosphoribosyl transferase, CaFur1p. Inactivation of these enzymes causes increased 5FC resistance [117].

Clinical isolates can develop resistance to multiple antifungals, as previously described in clinical isolates of *C. albicans*, *C. glabrata*, and *C. auris* [25,72,118]. Resistance to antifungals can be intrinsic (primary resistance) or acquired (secondary resistance) [119]. Intrinsic resistance is highly stable and is predictive of therapeutic failure. For example, *C. krusei* is intrinsically resistant to fluconazole. Further, intrinsic resistance to several antifungals including fluconazole and polyenes are also observed in *C. auris*, an emerging pathogen [119]. Recently, *C. auris* has proliferated at an alarming rate, causing severe problems in hospital settings. Mortality rates range from 30% to 60% and infections occur within weeks after hospital admission. Infection control is quite difficult because of high rates of colonization and environmental contamination by *C. auris* [119].

Besides intrinsic resistance, resistance to antifungals can be acquired. Acquired resistance can be either stable or transient. The acquired resistance develops due to long exposure of antifungals. For example, in studies with matched clinical isolates, where one set of *C. albicans* was isolated before azole treatment and one set after 6 months of azole treatment, *C. albicans* acquired azole resistance after azole therapy [25]. These resistance mechanisms were highly stable even after removal of azole pressure. Additionally, drug-induced aneuploidy can provide fitness advantages, as observed in *C. albicans* [120]. Aneuploidy, as described previously in this review, plays an important role in acquired drug resistance. These aneuploidies can be stable for several passages in the absence of antifungals, as observed in previous studies [104,120].

## 5. Conclusions

*Candidiasis* is caused by pathogenic yeasts, *Candida* sp., which are opportunistic pathogens present in the normal microbiome. The pathogens’ growth in the microbiome are checked by the host’s immune system. However, these pathogens overgrow in immune-compromised individuals, causing disseminated infections. Yeasts cannot be avoided in the day-to-day diet and some yeasts are useful in maintaining the host microbiome. Hence, total elimination of yeast from the body is neither desirable nor feasible. These infections are treated by different classes of antifungals that include azoles, polyenes, and echinocandins. These drugs target different biochemical pathways that are unique to yeasts. For example, azoles target the biosynthesis of ergosterol, a unique cell membrane component, present only in fungi. These antifungals are either fungistatic (azoles) or fungicidal (echinocandins). The efficacies of these drugs depend on the quality of the hosts’ immune system, site and severity of infection, and pharmacokinetics of the drugs. Besides drug efficacies, therapeutic failures may result from evolution of drug resistance in the infecting *Candida* sp. The molecular mechanisms associated with antifungal resistance were summarized in this review. These mechanisms include overexpression of membrane transporters, altered cell wall and ergosterol biosynthesis, gain of function mutations in the transcription factors regulating membrane transporters, and ergosterol biosynthesis. The ongoing research on better understanding of these mechanisms may aid in detecting resistant isolates, identify novel drug targets, and inhibiting the rise of drug resistance.

## Figures and Tables

**Figure 1 antibiotics-09-00312-f001:**
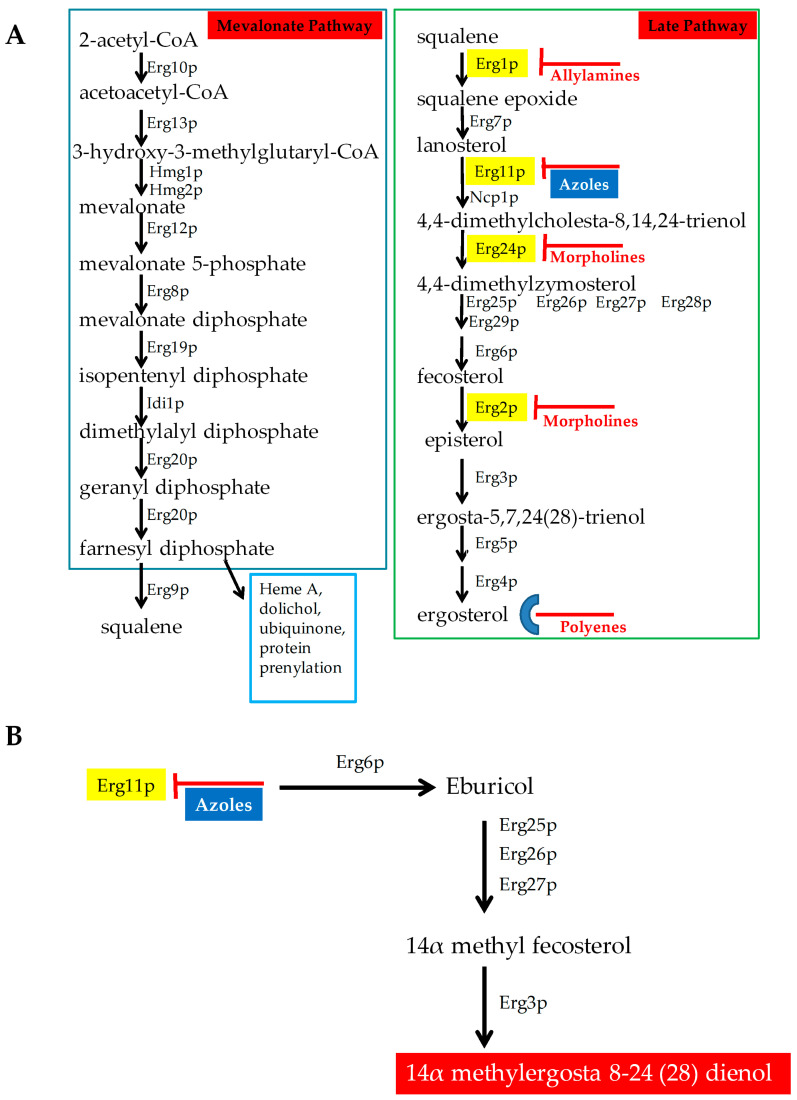
Antifungals that target ergosterol and its biosynthesis. (**A**) Ergosterol biosynthetic Pathway with different antifungals targeting different enzymes in the pathway. (**B**) Mechanism of action of azole antifungals.

**Figure 2 antibiotics-09-00312-f002:**
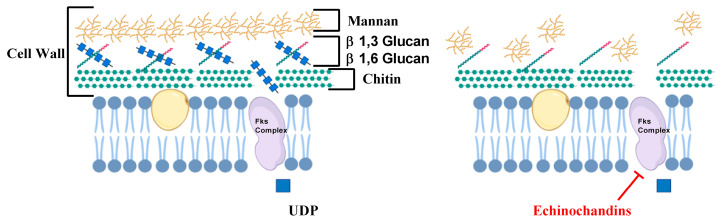
Role of echinochandin in inhibiting cell wall biosynthesis. Fks complex comprises of three proteins Fks1p, Fks2p and Fks3p that utilizes UDP-glucose (UDP) to synthesize β1-3 glucan. (Images were created with the help of biorender.com).

**Figure 3 antibiotics-09-00312-f003:**
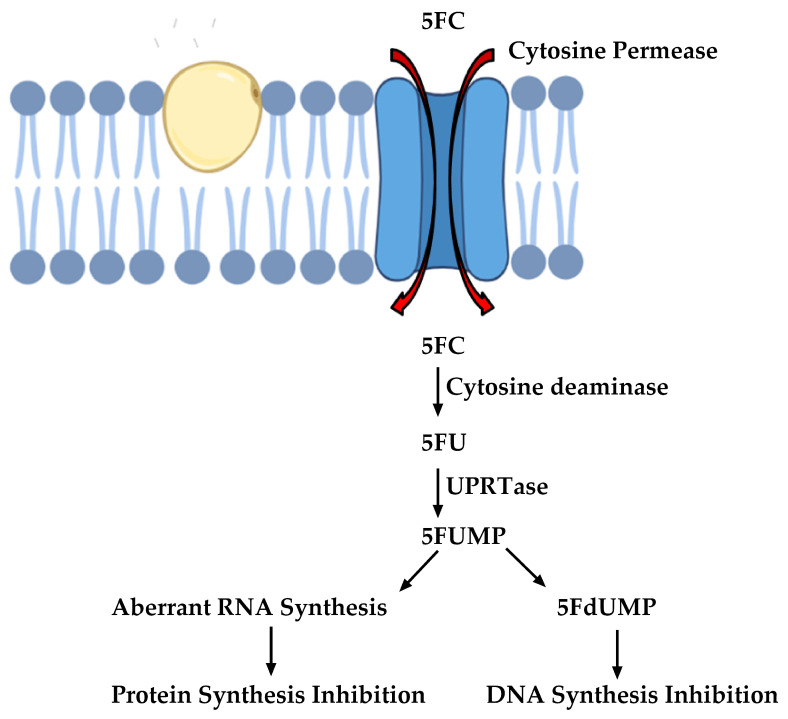
Mode of action of 5-Flucytosine (5FC) (Images were created with the help of biorender.com).

**Figure 4 antibiotics-09-00312-f004:**
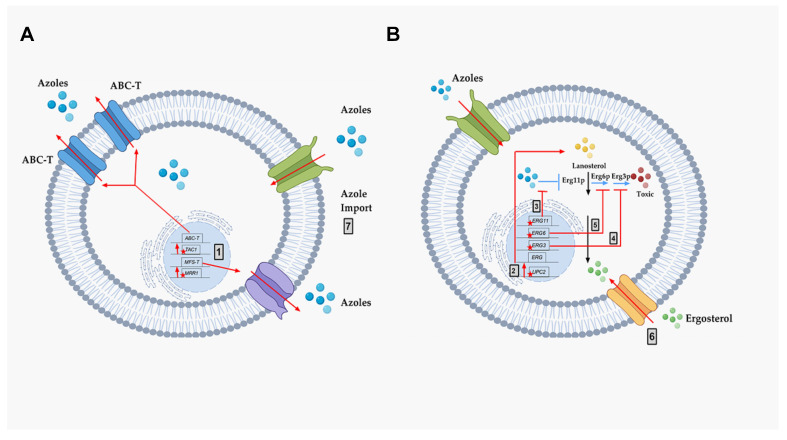
The molecular mechanisms of azole resistance. A schematic representation of known molecular mechanisms of azole resistance. Black arrows represent the normal ergosterol pathway, including lanosterol (Yellow Circles), ergosterol (Green Circles), and ergosterol inhibition of *UPC2*. Blue arrows represent the mechanism of action of azole drugs (Blue Circles), including the inhibition of target Erg11p, and production of toxic diol (Red Circles) through several steps, including Erg6p and Erg3p. The genes important for resistance are diagrammed within the nucleus as black rectangles in their chromosomal locations (black line). Red stars (mutations) and red arrows represent the different known molecular mechanisms of drug resistance in pathogenic fungi: (**A**) Azole resistance mechanism via efflux pump activities. (**B**) Azole resistance mechanism via altered ergosterol biosynthesis. Grey box 1—Increased efflux of azoles due to point mutations in *MRR1* and *TAC1* (stars), resulting in over-expression of MFS-Ts and ABC-Ts, respectively. (**B**) Grey box 2—Point mutations in *UPC2* causing increased expression throughout the ergosterol (*ERG*) pathway. Grey box 3—Point mutations in *ERG11* that prevent azole binding. Grey box 4—Point mutations in *ERG3* that prevent the formation of toxic sterol. Grey box 5—Point mutations in *ERG6* that prevent the formation of toxic sterol. Grey box 6—Sterol import that reduces the need for sterol biosynthesis. Grey box 7—Altered azole import that reduces intracellular azoles. This figure was created with the help of biorender.com.

**Table 1 antibiotics-09-00312-t001:** List of known multi-drug resistance-causing efflux pumps.

Efflux Pump	Pump Type	Organism
Cdr1p, Cdr2p	ABC-T	*Candida albicans*
CgCdr1p	ABC-T	*Candida glabrata*
CgFlr1p	MFS-T	*Candida glabrata*
CgPdh1p	ABC-T	*Candida glabrata*
CgQdr2p	MFS-T	*Candida glabrata*
CgSnq2p	ABC-T	*Candida glabrata*
CkAbc1p	ABC-T	*Candida krusei*
CkAbc2p	ABC-T	*Candida krusei*
Mdr1p	MFS-T	*Candida albicans*
Cdr1p	ABC-T	*Candida auris*

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
