# Peer review of "Candidiasis and Mechanisms of Antifungal Resistance"

_antibiotics, 2020, doi:10.3390/antibiotics9060312_

Round 1

Reviewer 1 Report

The manuscript is written in good English, it is well-structured, and very clear for a reader. This review paper contains 90 references and sufficiently describes the mechanisms of antifungal resistance of invasive Candida.

  1. It is necessary to redraw Figure 1 to make it more accurate and clear for the readers. For example, see a figure here: https://www.cmaj.ca/content/180/4/408/F3
  2. Figure 2 should be re-organized (divided to a few?) to be more clear. It looks overloaded now.
  3. The sub-section 4.1.5 "Genome Plasticity" should be extended. It is too short now and does not describe all paths.

Author Response

Please See the attachment in the box

Reviewer 2 Report

Review of the manuscript entitled “Candidiasis and mechanisms of Antifungal Resistance

This work is a review. Its purpose was to provide an overview about the molecular mechanisms of action of antifungal as well as mechanisms of drug resistance used by Candida sp.

The review is quite interesting, fully documented and exhaustive, especially because it is not limited to Candida albicansand extends to other emerging Candida species. However, this review needs to be improved, especially concerning some specific points (see details below) and illustrations that are rather sloppy or not very informative (which is a pity given the good quality of the overall text).

  1. L39-41. Rather elusive; please could you specify the average proportions in which these different species are recovered, especially the most important one, C. albicans, compared to other Candida but also for C. auris which has been more recently described. Have these proportions been stable in recent years, or is there a clear increase in some emerging species, to which special attention should be paid?

  1. The illustrations are very disappointing. The referee thinks that the Figure 1 “Different Types of Antifungals and Their Targets” is of no interest at all because it is very schematic, not very detailed and not informative. It is mentioned in the text in the paragraph that details the ergosterol biosynthesis pathway, which is quite curious ... but here, we are missing a figure of this biosynthesis pathway with the 25 genes mentioned. The reviewer knows that ergosterol pathway is illustrated (again very schematically, and not very clearly) later in the manuscript (figure 2, azole resistance) but there it is sorely missing for a good understanding. It could be of great interest to add a figure including the most commonly used molecules in each family, which, for example, would allow a better understanding of the difference between imidazoles and triazoles. An illustration about echinocandins antifungal drugs target should be also added.

  1. L119-137, it's very interesting (and not so common!) to cite other antifungal molecules that are effective but not used to fight candidiasis. Examples in human health are given. Are there other classes of antifungal molecules used on phytopathogenic fungi that are faced to problems of resistance?

  1. Concerning azole drug resistance, are the different mechanisms presented in order of importance? This needs to be specified and even developed a little: e.g., if efflux pumps are in the majority in the described cases of candidiasis resistance, what proportion do they represent?

  1. paragraph “4.1.4 Altered Azole Import”. It seems that this mechanism of resistance has never been reported yet? The reviewer is not convinced that this mechanism, which remains very hypothetical, should be indicated in the same way as other proven mechanisms. Maybe in this case, make a paragraph on other putative resistance mechanisms not described in Candida or other related fungi, such as Azole import or detoxification?

  1. Overall, the different resistance mechanisms are presented one after the other and according to the classes of molecules. However, different cases of Candida isolates accumulating multi-resistance to different antifungal agents according to different resistance mechanisms have been reported (as in Chapeland-Leclerc et al. 2010, Antimicrobial Agents and Chemotherapy 54(3):1360-2). Moreover, if these different isolates (mono or multi-resistant) remain relatively rare or proliferate in a moderate way, it may be because they are not really very competitive, considering their fitness, and disappear when there is no more selection pressure. It seems to the referee that these notions should be discussed in a separate paragraph.

Reviewer 3 Report

In this review entitled Candidiasis and mechanisms of antifungal resistance, authors have described briefly the available anti-fungal agents against Candidiasis and overviewed various antifungal resistance mechanisms of Candida sp.

Please check for the minor grammars and proofread.

Reviewer 4 Report

This review describes the main antifungal drug classes used in clinical practice, their modes of action, and the molecular resistance mechanisms developed by Candida spp. The manuscript could provide a broader view in some sections where abundant evidence is available, thus further increasing the impact of the review.

Major comments:

1. I have detected several instances where references are used incorrectly or misplaced. Since this is a review manuscript, I recommend the authors to be careful when exposing information from other studies and cautiously analyze the data they are referencing. The manuscript should carefully convey the actual data from the cited studies instead of transmitting it in a simplistic way, which results in erroneous information in the manuscript. Some examples:

-Lines 250-252: The cited test relates to the role of Mrr1 in C. dubliniensis azole resistance (with only some heterologous expression experiments using C. albicans MRR1). Also, there is no reference to the mutation P683H in the study. Alternatively, the authors may want to reference the following study: doi: 10.1371/journal.ppat.0030164.

-Lines 258-262: The cited studies do not provide the information conveyed in the text. One study shows Yap1 regulation over Flr1 under benomyl stress and not transporter overexpression during azole stress. Plus, the strains used in the cited studies are not azole resistant and I do not believe they are clinical isolates. However, MFS transporters were indeed shown to be overexpressed in C. glabrata clinical isolates (doi: 10.3389/fmicb.2016.00526).

-Lines 264-265: The cited study provides evidence of Pdr1 regulation over QDR2 expression, however there is no data relative to Pdr1 expression or protein activity.

-Lines 322-323: "Upc2p is a Zn2-Cys6 zinc cluster transcription factor, similar to CaTac1p and CaMrr1p". I believe the authors meant that Upc2 is a Zn2-Cys6 transcription factor, as are Tac1 and Mrr1. However, the rest of their protein sequences share no similarity. Further, referencing the study that first described Upc2 seems misleading, as there is no comparison with Tac1 and Mrr1 in that study.

-Lines 355-356: Reference 81 refers to Aspergillus fumigatus, not C. albicans. The study references azole import studies in C. albicans, but it is not the primary reference. The authors should always reference the original study from where the data comes from.

2. Lines 217-225: It is stated that ABC transporters require an energy source and MFS transporters require a proton gradient. This is obviously true, but the phrasing appears redundant. Both ABC and MFS transporters require an energy source, just different sources. This makes ABC transporters part of the primary transport system, while MFS transporters perform secondary transport. Further, I caution that MDR transporters are capable of altering the intra/extracellular distribution a wide variety of unrelated compounds (among them, but not exclusively, antifungal drugs). The text reads "Azole are the substrates...", but azoles are not the only substrates recognized by these transporters; most of them are involved in the transport of physiological metabolites which are possibly the "original" substrates. Moreover, although some pumps have been shown to directly transport antifungal drugs, increasing evidence has shown that differential accumulation of xenobiotics can be an indirect effect due to transport and homeostasis of other substrates. I strongly recommend the authors to search the available literature and further develop this section of the manuscript to accurately introduce the multifactorial nature of membrane transporters and its consequences in antifungal resistance.

3. Lines 263-265: It would help to provide more information on the role of the TF Pdr1 in C. glabrata. Pdr1 is the master regulator of azole resistance and, to date, Pdr1 GOF mutations is the only resistance mechanism unequivocally associated with clinical acquisition of azole resistance in this species. There is significant literature available to review on this topic.

4. Lines 176-179: There is no mention of side-effects in the text. All classes of antifungal drugs have well-established side-effects that should be discussed in this section.

Other comments:

5. Page 4: polyene description is under section 2.1 Inhibitors to ergosterol biosynthesis. As stated in the text, polyenes target ergosterol directly and not its biosynthetic pathway. It could be best to create a sub-section or rename the 2.1 section to better describe that not all agents target ergosterol biosynthesis. The same is valid for the text in lines 85-86.

6. It should be stated that polyenes exert fungicidal action.

7. Lines 95-96: It should be included that the referred systemic triazole agents are both more potent and less toxic than, for example, fluconazole. I also suggest to include references supporting this statement.

8. Lines 136-137: Echinocandins target the product of the FKS1 gene, but FKS1 only encodes a component of the beta-1,3-glucan synthase enzyme. This should be more clearly stated. Further, it would be relevant to mention that echinocandins are used especially in patients with systemic infections with previous azole exposure.

9. Lines 170-172: How does the different immune response in distinct niches correlate with drug efficacy? It would add to the manuscript to provide more detailed information on current evidence of this or mechanistic insights if they exist.

10. Figure 2 legend: The numbers from Grey box 6 and 7 are missing in the legend text. Grey box 7 legend reads "Altered azole import that reduced intercellular azoles". Do the authors mean intracellular azoles? If not, specify the meaning of intercellular azoles.

11. Lines 377-383: The text should feature that mutations in hotspot regions in C. glabrata FKS2 have also been identified, with relevant outcomes in clinical echinocandin resistance.

12. Lines 394-395: For clarity, consider changing to "causing disseminated infections", as simply "infections" can be caused in the so-called healthy individuals, not only immuno-compromised patients.

Round 2

Reviewer 4 Report

The authors have appropriately addressed the concerns raised by the reviewer.

Author Response

Thank You for the wonderful comment